# Multi-Marker Immunofluorescent Staining and PD-L1 Detection on Circulating Tumour Cells from Ovarian Cancer Patients

**DOI:** 10.3390/cancers13246225

**Published:** 2021-12-10

**Authors:** Du-Bois Asante, Michael Morici, Ganendra R. K. A. Mohan, Emmanuel Acheampong, Isaac Spencer, Weitao Lin, Paula van Miert, Samantha Gibson, Aaron B. Beasley, Melanie Ziman, Leslie Calapre, Tarek M. Meniawy, Elin S. Gray

**Affiliations:** 1School of Medical and Health Sciences, Edith Cowan University, Perth, WA 6027, Australia; dasante@our.ecu.edu.au (D.-B.A.); m.morici@ecu.edu.au (M.M.); eacheam1@our.ecu.edu.au (E.A.); i.spencer@hotmail.co.uk (I.S.); w.lin@ecu.edu.au (W.L.); ppmcvanmiert@gmail.com (P.v.M.); a.beasley@ecu.edu.au (A.B.B.); m.ziman@ecu.edu.au (M.Z.); l.calapre@ecu.edu.au (L.C.); Tarek.Meniawy@health.wa.gov.au (T.M.M.); 2Centre for Precision Health, Edith Cowan University, Perth, WA 6027, Australia; 3Hollywood Private Hospital, Perth, WA 6009, Australia; raj.ganendra@gmail.com; 4Harry Perkins Institute of Medical Research, Perth, WA 6009, Australia; 5Western Oncology, Perth, WA 6008, Australia; samgibson@westernoncology.com.au; 6School of Biomedical Science, University of Western Australia, Perth, WA 6009, Australia; 7School of Medicine, University of Western Australia, Perth, WA 6009, Australia; 8Department of Medical Oncology, Sir Charles Gairdner Hospital, Perth, WA 6009, Australia

**Keywords:** high grade serous ovarian cancer, HGSOC, immunofluorescence, circulating tumour cells, CTC, fluorescence quenching, mesenchymal, epithelial

## Abstract

**Simple Summary:**

Circulating tumour cells (CTCs) have the potential to serve as a rich source of information for cancer diagnostic and therapeutic decisions. To fully exploit this minimally invasive diagnostic resource requires techniques that aid in enriching heterogenous populations of CTCs and markers to efficiently characterise these cells as tumour derived. In the present study we eva-luated the microfluidic enrichment of CTCs and a multi-marker staining methodology for the identification of heterogeneous CTCs in ovarian cancer (OC) patients and evaluation of PD-L1 expression. We showed, for the first time, the existence of hybrid CTCs with an epithelial/mesenchymal phenotype and their association with PD-L1 in OC. Incorporation of this method in future clinical trials can help predict immunotherapy responsiveness in OC patients.

**Abstract:**

Detection of ovarian cancer (OC) circulating tumour cells (CTCs) is primarily based on targeting epithelial markers, thus failing to detect mesenchymal tumour cells. More importantly, the immune checkpoint inhibitor marker PD-L1 has not been demonstrated on CTCs from OC patients. An antibody staining protocol was developed and tested using SKOV-3 and OVCA432 OC cell lines. We targeted epithelial (cytokeratin (CK) and EpCAM), mesenchymal (vimentin), and OC-specific (PAX8) markers for detection of CTCs, and CD45/16 and CD31 were used for the exclusion of white blood and vascular endothelial cells, respectively. PD-L1 was used for CTC characterisation. CTCs were enriched using the Parsortix™ system from 16 OC patients. Results revealed the presence of CTCs in 10 (63%) cases. CTCs were heterogeneous, with 113/157 (72%) cells positive for CK/EpCAM (epithelial marker), 58/157 (37%) positive for vimentin (mesenchymal marker), and 17/157 (11%) for both (hybrid). PAX8 was only found in 11/157 (7%) CTCs. In addition, 62/157 (39%) CTCs were positive for PD-L1. Positivity for PD-L1 was significantly associated with the hybrid phenotype when compared with the epithelial (*p* = 0.007) and mesenchymal (*p* = 0.0009) expressing CTCs. Characterisation of CTC phenotypes in relation to clinical outcomes is needed to provide insight into the role that epithelial to mesenchymal plasticity plays in OC and its relationship with PD-L1.

## 1. Introduction

Circulating tumour cells (CTCs) are rare neoplastic cells found in the circulatory system, thought to be shed from primary, metastatic, or recurrent tumours. CTCs, or a proportion of them, can form micro-metastases in other organs beyond the original tumour site [1,2]. Their rarity in whole blood means that efficient methods for both enrichment and detection of CTCs are needed. The advent of new and user-friendly technologies is crucial to advancing the application of CTCs as a liquid biopsy for precision medicine [3].

The U.S. Food and Drug Administration (FDA)-approved CellSearch^®^ system uses immunoaffinity for the capture of CTCs that express the epithelial cell adhesion molecule (EpCAM) [4]. As a result, this method may fail to isolate CTCs with a mesenchymal phenotype [5], thought to be more metastatic, and have been reported to be associated with therapy resistance and poor patient outcomes [6,7]. Epitope-independent isolation platforms have become more prevalent in recent years as they allow for enrichment of a broad spectrum of CTCs that may not express cognate antigens [3]. For example, the Parsortix™ microfluidic system isolates rare cells from bodily fluids based on cell size and compressibility [8,9].

Traditionally, detection of CTCs is restricted to 3–4 fluorescent channels using fluorescence microscopy. Consequently, CTCs are identified as nucleated cells (DAPI/Hoechst), positive for the epithelial markers EpCAM and cytokeratin, and negative for the leucocytic marker CD45 as exemplified by the CellSearch^®^ protocol [4]. This limits the phenotypic characterisation of detected CTCs. CTCs can be phenotypically and genotypically heterogeneous, with each cell detected, representing characteristics of the tumour of origin [10]. Because of intra- and inter-tumoural heterogeneity, CTCs from cancer patients may carry epithelial traits, which can be detected by EpCAM and cytokeratin, or mesenchymal traits such as N-cadherin and vimentin [7]. Additionally, these two phenotypes are generally mutually exclusive [11]. Thus, using only epithelial markers for staining may fail to identify mesenchymal CTCs [12]. For instance, in a recent ovarian cancer (OC) study, adding anti-N-cadherin to anti-EpCAM antibodies enhanced the isolation and subsequent detection of mesenchymal CTCs [13].

A multi-marker subtyping of CTCs employing sequential fluorescent quenching and re-staining (FQR) of isolated CTCs on CellSieve^®^ has been reported recently. The researchers used a combination of borohydride (BH_4_) and Tris solution as quenching agents to reduce fluorescence to a dark state, allowing subsequent fluorescence staining targeting different markers for a second [7] and third time [14]. This method allows for the detection of phenotypically heterogeneous subpopulations of rare cells and thereby, screening of multiple clinically actionable biomarkers on captured CTCs. However, although this technique is inexpensive and simple, it is time consuming, and CTCs need to be captured on specialised CellSieve^®^ as opposed to the conventional glass slides that are usually used in routine clinical laboratories.

Ovarian cancer (OC) is histologically heterogeneous, with epithelial ovarian carcinoma making up about 90% of all cases of the disease. Epithelial ovarian carcinoma consists of four distinct histological subtypes, namely, clear cell, serous, mucinous, and endometrioid. The high grade serous ovarian carcinoma (HGSOC) subtype is the most clinically aggressive and common form of the disease [15]. Thus, techniques that enhance efficient detection of CTCs with markers of biological importance in this lethal OC subtype are warranted.

Here we describe a methodology that affords multiple rounds of FQR on enriched cells on a glass slide in a single day protocol. Using this protocol, we identified CTCs isolated using the Parsortix™ cell separation system from predominantly HGSOC patients. Moreover, we evaluated the expression of epithelial and mesenchymal markers, as well as PD-L1 on the isolated CTCs.

## 2. Materials and Methods

### 2.1. Cell Lines

The OC cell lines SKOV-3 and OVCA432 [16] were provided by Prof. Sai Wah Tsao (Li Ka Shing Faculty of Medicine, Pokfulam, HK). MCF-7 (breast cancer cell line) was obtained from the Melanoma Research Group, Edith Cowan University, WA, Australia. Cell lines were cultured in RPMI-1640 medium (Thermo Fisher Scientific, Waltham, MA, USA), supplemented with 5% foetal bovine serum (FBS) (Thermo Fisher Scientific).

The cultures were maintained at 37 °C in a humidified atmosphere containing 5% (*v/v*) CO_2_ until 80% confluence. Cell lines were trypsinised using 0.5% trypsin-EDTA (Thermo Fisher Scientific) diluted to 0.05% with 1X phosphate-buffered saline (PBS) solution (Thermo Fisher Scientific). The mean cell diameter of the two cell lines was measured using the Countess^®^ II FL automated cell counter (Thermo Fisher Scientific).

### 2.2. Optimisation of Antibodies for Immunostaining

For optimisation of antibodies, cell lines were cultured for 24 h on coverslips in 6-well plates (5 to 7 passages were done after thawing from liquid nitrogen before growing cells on the coverslip). Cells were washed with PBS and immediately fixed with 4% paraformaldehyde for 10 min. Cells were then permeabilised and blocked with antibody buffer containing 3% Bovine Serum Albumin (BSA) (Bovogen Biologicals Pty Ltd., East Keilor, VIC, Australia), 0.2% Triton X-100 (Sigma Aldrich, St. Louis, MO, USA) in PBS, 1 M glycine, 10% normal donkey serum (NDS) (Sigma Aldrich), and FcR blocking reagent (Miltenyi Biotech, Bergisch Gladbach, Germany) for 15 min at room temperature. The following antibodies were tested and optimised individually on the cell lines in a staining buffer containing 10% NDS, 3% BSA, 0.2% Triton X-100 PBS solution: FITC-labelled anti-Cytokeratin (1:50) (CK3-6H5, Miltenyi Biotec), FITC-labelled anti-EpCAM (1:100) (VU-1D9, Thermo Fisher Scientific), AlexaFluor 647-labelled anti-PD-L1 (1:100) (clone 28-8, Abcam, Cambridge, UK), AlexaFluor 488-labelled anti-PAX8 (1:100) (clone EPR18715, Abcam) and AlexaFluor 647-labelled anti-vimentin (1:1000) (clone V9, Abcam). Anti-PAX8 is an ovarian cancer-specific antibody with a nuclear staining pattern that has been used to stain carcinomas of the ovary [17,18]. Optimisation of the nuclear stain (PAX8 antibody) was done using SKOV-3 as a positive cell line and MCF-7 as the negative control (Appendix A).

For testing PD-L1 expression, the cell lines (SKOV-3 and OVCA432) were incubated with 100 ng/mL interferon-gamma (IFN-γ) (Shenandoah Biotechnology, Warwick, PA, USA) for 24 h to induce the expression of PD-L1 [19]. Additionally, we employed a combination of PE-labelled anti-CD45 (1:50) (clone HI30, BD Biosciences, San Jose, CA, USA), the traditional negative selection marker, and PE-labelled anti-CD16 (1:50) (clone 3G8, Biolegend, San Diego, CA, USA) for white blood cell (WBC) staining. The additive effect of these two markers improved the staining coverage of WBCs [20] compared to their individual use (Appendix A). PE-labelled anti-CD31 (1:50) (clone WM59, BD Biosciences) was also used to exclude vascular endothelial cells in the enriched samples.

### 2.3. Spiking Experiment and Enrichment Platform

We tested the recovery rate (capture efficiency) of the OC cell lines (SKOV-3 and OVCA432) spiked into 7.5 mL fresh blood from female healthy control subjects using the semi-automated microfluidic device Parsortix™ (Angle plc, Guildford, Surrey, UK). The separation was run at the default pressure setting of 99 mbar. The isolation of the spiked cell lines in the blood occurred in a disposable microscope slide-sized cassette with a final separation size of 6.5 µm. Firstly, 50 cells (each from SKOV-3 and OVCA432) were harvested from culture flasks using 0.5% trypsin-EDTA and were manually counted and directly spiked into fresh whole blood samples from healthy female volunteers. The spiking experiment was carried out in duplicate for both cell lines. After enrichment with the Parsortix™, captured cells were re-suspended in PBS containing 2% paraformaldehyde (fixation) for 10 min, cytospun (Cytospin™ 4, Thermo Fisher Scientific) onto glass slides (Shandon™ Single Cytoslides™, Thermo Fisher Scientific) at a speed of 2000 rpm for 5 min. The glass slides were immune-stained, and the recovery rate of cells determined.

### 2.4. Immunostaining of Spiked Cells and CTCs on Glass Slides Using the Fluorescent Quenching and Re-Staining Method

We employed the sequential FQR method adapted from Adams et al. [14,21] for detection of spiked cell lines and subsequently extended this approach for the staining of CTCs isolated from HGSOC patient samples. Briefly, enriched cells on glass slides were first permeabilised and blocked with 3% BSA, 0.2% Triton X-100, PBS, 1 M glycine, 10% NDS, FcR blocking reagent solution for 15 min at room temperature. Cells were then incubated for 60 min with the first antibody panel consisting of FITC-labelled anti-Cytokeratin and anti-EpCAM, Phycoerythrin (PE)-labelled anti-CD45 and anti-CD16, and Alexa Fluor 647-labelled anti-PD-L1. The antibodies were removed and 2 µg/mL of the nuclei staining dye, Hoechst 33342 (Thermo Fisher Scientific), was added for 15 min. Cells were washed with 1% BSA-PBS and then finally with PBS.

For fluorescent imaging, a wet mounting approach was used instead of the traditional resinous/adhesive mountants (Appendix A). Briefly, a silicone isolator was immediately placed on the glass slide encircling the area marked on the slide where the cells were located. PBS (200 µL) was put in the circular space containing the cells, enabling immediate or same-day imaging of stained slides. Cells were immediately visualised and scanned using the inverted fluorescent microscope (Eclipse Ti-E, Nikon^®^, Tokyo, Japan), and analysed using the NIS-Elements High Content Analysis software (version 4.2).

After microscopy, the silicone isolator was removed and fluorescent markers on the cells quenched using 1 mg/mL BH_4_ solution followed by 100 mM Tris (pH = 9.0) as reported earlier [21] using the FQR method, with few modifications. Briefly, the silicone isolator was removed, and the glass slides were incubated (room temperature) in freshly prepared BH_4_ solution for 90 min (two times). Slides were then washed in changes of PBS (5 min, six times). It was then incubated in Tris solution for 60 min and washed in changes of PBS (2 min, three times). Cells were re-stained using the second antibody panel comprising Alexa Fluor 488-labelled anti-PAX8, PE-labelled CD31 and Alexa Fluor 647-labelled anti-vimentin in the staining solution. The silicone isolator was placed on the slide and PBS added as done earlier, and the slide re-imaged.

During fluorescent imaging, x/y placement of cells was recorded in the first staining panel to aid the relocation of previously imaged cells in the second staining after quenching. Morphologically intact nucleated cells (Hoechst^pos^) that are CK^pos^/EpCAM^pos^/vimentin^pos^/PAX8^pos^, with or without PD-L1 expression, and CD45^neg^/CD16^neg^/CD31^neg^ were interpreted as recovered cell lines or CTCs. The SKOV-3 cell line was used for the optimisation of the quenching method (Appendix A). For final image processing of the first and second staining, ImageJ (Version 1.52a, National Institutes of Health, Bethesda, MD, USA) was used to overlay the individual channels into a final merged image of cell lines and CTCs.

After re-imaging, the slides were mounted with Fluoromount Gold^TM^ with DAPI (Thermo Fisher Scientific), covered with a coverslip, and stored at 4 °C for future use.

### 2.5. Isolation and Detection of CTCs from the Blood of Patients and Healthy Controls

A total of 16 whole blood samples from 16 OC patients were collected for this study. Fifteen (15) were of the HGSOC subtype, while one patient had mucinous borderline (Table 1). Five (5) healthy female volunteers (age: 28–55) donated blood for the study. Both healthy volunteers and HGSOC patients signed consent forms approved by the Human Research Ethics Committees at Edith Cowan University (No. 18957) and Sir Charles Gairdner Hospital (No. 2013-246 and RGS0000003289). For each participant, an SST (BD Biosciences) was used to draw approximately 4 mL of blood as a discard to avoid contamination from epithelial and endothelial cells from the skin and blood vessels, respectively. Then, 7 to 10 mL of whole blood was collected in BD Vacutainer K2 EDTA tubes (BD Biosciences). Blood samples were processed within 6 h post collection. CTCs were enriched from whole blood samples using the Parsortix™ and subsequently stained using the immunostaining protocol described above.

The total number of WBCs (CD16^pos^/45^pos^) in the Parsortix™-enriched cell fractions were quantified in samples from healthy volunteers and patients (Appendix A).

### 2.6. Statistical Analysis

Cell sizes were compared using Student’s *t*-test. The frequency of PD-L1-expressing CTCs across different subpopulations was compared using Fisher’s exact test. All graphs and statistical analyses were done using the GraphPad Prism (version 6.01, San Diego, CA, USA).

## 3. Results

### 3.1. Phenotypic Characterisation of OC Cell Lines

Immunocytochemistry (ICC) staining of the OC cell lines SKOV-3 and OVCA432 cultured on coverslips showed differential expression of EpCAM, CK, and vimentin (Figure 1). Both cell lines were positive for the OC-specific nuclei marker PAX-8. The more cluster forming cell line, OVCA432, showed high expression of the epithelial markers EpCAM and CK, while the less cluster forming and larger cell line, SKOV-3, generally demonstrated very weak expression for EpCAM and strong expression of the mesenchymal marker vimentin [22]. CK expression, however, was present in a few highly stained cells studded among numerous weakly stained SKOV-3 cells. Thus, the ICC staining categorised the two OC cell lines into a more epithelial (OVCA432) and a more mesenchymal (SKOV-3) phenotype.

Following 24-h induction with IFN-γ, SKOV-3 cells also demonstrated moderate expression of PD-L1. However, OVCA432 cells showed very weak to negative expression of PD-L1, even in the presence of IFN-γ. These results suggested a high association of PD-L1 expression to the more mesenchymal cell line, SKOV-3, in agreement with an earlier report [23]. On the other hand, prior information on PD-L1 expression post-IFN-γ induction in OVCA432 was lacking.

### 3.2. Immunocytochemistry of Spiked Cell Lines

We spiked the two OC cell lines into 7.5 mL of whole blood from healthy female individuals, enriched with the Parsortix™ system and subsequently detected by ICC. Spiked cell lines (Figure 2) demonstrated similar staining patterns as those cultured and stained on the coverslip (Figure 1). SKOV-3 was found to have moderate to low expression of the epithelial markers (CK/EpCAM), moderate expression of PD-L1, and high expression of the mesenchymal marker (vimentin). By contrast, OVCA432 showed high expression of CK/EpCAM and a lack of PD-L1 and vimentin expression. Both cell lines, however, showed moderate to high expression of nuclear marker PAX8.

Of note, the microfluidic device isolated clusters of cells spiked into the healthy control blood (Figure 2). Overall, the combination of our staining protocol and the use of the microfluidic device appeared to be useful in isolating and detecting single or clustered cells; this method could potentially be extended to the enrichment and detection of these heterogeneous subpopulations of cells in patient blood samples.

### 3.3. Recovery Rate of Spiked Cells

To determine the efficiency of the microfluidic Parsortix™ device to isolate OC CTCs of different cell sizes, blood samples from healthy controls were spiked with 50 cells (each from SKOV-3 and OVCA432). The spiking experiment was carried out in duplicate for both cell lines. Following enrichment with the Parsortix™ system, cells were then cytospun onto the glass slides and stained. In addition, blood samples (no spike) from three healthy females were used as negative controls. The control together with the spiked samples were stained and analysed. The healthy control slides were negative for the tumour markers (CK/EpCAM, PAX8 and vimentin), but a few WBCs showed co-expression of PD-L1 and CD16/45.

SKOV-3 and OVCA432 had a mean recovery rate of 89 and 71%, respectively, depicting a lower capture rate for OVCA432 relative to SKOV-3 cells. Comparing the mean diameters of the two cell lines (*n* = 10), OVCA432 was considered the smallest with a mean cell diameter of 13 μm versus 17 μm for SKOV-3. The mean cell diameter significantly differed between these two OC cell lines (*p* = 0.0019) (Figure 3). The difference in the cell diameter combined with deformability could account for the difference in the recovery rates observed earlier. Obermayr et al. worked with a final separating cassette gap size of 10 µm, which will not be able to capture as many spiked cells [24]. Alternative cassette sizes (6.5 and 8 µm) have been developed by the manufacturer (Angle plc). Comparing the 6.5 and 10 µm cassettes, the 6.5 µm demonstrated a significantly higher capture rate than the latter when spiked in whole blood in EDTA tubes (discussed in [25]). We used a 6.5 µm final cassette gap size, which had a higher capture rate (~66–92%) [8], thus our results.

### 3.4. CTC Detection from Patient Blood

Following successful immunostaining of the OC cell lines, the methodology was applied to the analysis of blood samples obtained from OC patients. We enrolled a total of 16 women with OC, mostly newly diagnosed. Their characteristics are shown in Table 1. The median age at diagnosis was 68 years (range 50–83 years). All but one patient had a HGSOC histology, with one patient having a mucinous borderline tumour. The FIGO stages were predominantly IIIC and IV, with two stage I (IA and IC). Ten (10) newly diagnosed OC patients had three cycles of platinum-taxane-based neoadjuvant chemotherapy (NACT), and the others underwent upfront debulking surgery. Interval debulking surgery and evaluation of chemotherapy response score (CRS) was performed for eight individuals. Three (3) of the NACT group (OC 1362, 1364, and 1458) did not undergo interval debulking surgery. OC 1362 and 1364 had progressive disease (PD) after 1st line NACT, while OC 1458, on the other hand, died before the completion of the NACT.

Our immunostaining method successfully aided in the identification of CTCs in 63% (10/16) of the patients, with numbers ranging from 2–49 CTCs (median = 4) in 8 mL of blood. A total of 157 CTCs were identified in the blood of the 10 CTC-positive cases. Table 2 summarises the CTCs detected using each of the markers.

Figure 4 shows the differential expression of markers from representative CTCs detected in these patients. CTCs were heterogenous, 113 cells were CK^pos^/EpCAM^pos^ and 58 cells were vimentin^pos^ (Table 2). Of these, 17 (11%) cells were positive for both epithelial and mesenchymal markers (Hybrid CTCs: H-CTCs), while 96 (61%) were exclusively epithelial CTCs (E-CTCs) and 41 (26%) mesenchymal CTCs (M-CTCs). Eleven cells expressed the marker PAX8, with only three (2%) cells expressing PAX8 alone. Samples from recurrence HGSOC patients (OC 714 and 1388) had more M-CTCs (4 and 6) as compared to E-CTCs (2 and 3) (Table 2). Samples from newly diagnosed patients, however, showed no preference for either M- or E-CTCs; some cases (OC 1362, 1364, and 1382) showed enrichment of predominantly E-CTCs, while others (OC 1251 and 1423) showed CTCs that were M-CTCs. In one case (OC 1423), we identified two CTC clusters, ranging from 6–11 cell per cluster. All the clusters were vimentin^pos^, belonging to the M-CTC phenotype (Figure 5).

### 3.5. Marker Co-Expression and Exclusion of CD31^pos^ Endothelial Cells

Double positive cells comprising CK^pos^/EpCAM^pos^ and CD16^pos^/45^pos^, vimentin^pos^ and CD16^pos^/45^pos^, CK^pos^/EpCAM^pos^ and CD31^pos^, or vimentin^pos^ and CD31^pos^ were not considered as CTCs, thus are not included in Table 2. The number of CD31^pos^ endothelial cells identified in OC patients and healthy volunteers and the marker they expressed are tabulated in Appendix A. Notably, expression of CK/EpCAM was identified in 13 (39%) of these cells. This finding was unexpected and highly relevant, as CD45^neg^/CK^pos^/EpCAM^pos^ cells are always defined as CTCs, and CD31 is uncommonly used as an exclusion marker in CTC studies.

The combined expression of both CK/EpCAM and vimentin (CTC markers) on the endothelial cells were seen in 22 (67%) (Appendix A) of the cells, thus reducing potential false positives of our CTC markers. Normally, classical vascular endothelial cells should have the expression of CD31 and vimentin (Figure 6a,b). In our study, most vascular endothelial cells identified from the OC patients that were CK/EpCAM and PD-L1 positive had no expression of vimentin (Figure 6c–e).

### 3.6. Comparing PD-L1 Expression on Mesenchymal and Epithelial CTCs

PD-L1 was predominantly expressed on single CTCs as shown in Figure 4, Figure 6 and Figure 7. Of the 157 CTCs detected in 10 OC patients, 62 were found to express PD-L1. Only five patients had PD-L1^pos^ CTCs (Table 2), representing 50% of the patients with detectable CTCs. PD-L1^pos^ CTCs predominantly showed moderate expression of the PD-L1 marker, whether membranous or cytoplasmic (Figure 7A).

We further analysed PD-L1 expression relative to its epithelial or mesenchymal phenotype (Figure 7B). PD-L1^pos^ CTCs accounted for 11/41 (27%) M-CTCs and 38/96 (40%) E-CTCs (Fisher’s exact test, *p* = 0.177). PD-L1^pos^ CTCs were more common among H-CTCs, representing 13/17 (76%) of these cells. Positivity for PD-L1 was significantly associated with H-CTCs when compared with epithelial (*p =* 0.007) and mesenchymal (*p =* 0.0009) CTCs (Fisher’s exact test). Overall, our results suggested an association between PD-L1 expression and a hybrid phenotype in OC-derived CTCs.

## 4. Discussion

The evolution of new technologies to aid efficient enrichment and subsequent detection of CTCs has been key to the recent advancement in the analysis of these rare blood cells [26]. Nonetheless, the lack of standardised protocols necessitates more robust and user-friendly technologies, which are paramount to galvanising CTC analysis into a reliable diagnostic and prognostic tool. This pilot study affirmed the potential use of label-free CTC enrichment and sequential multi-marker staining methods to effectively detect heterogeneous epithelial (CK/EpCAM^pos^) and mesenchymal (vimentin^pos^) CTCs in OC patients.

Assessment of markers for enumeration and phenotypic characterisation of CTCs derived from OC patients demonstrated that CTCs are indeed heterogeneous, bearing markers of either the epithelial (CK^pos^/EpCAM^pos^) [27,28,29,30] and/or mesenchymal [13,31] phenotype. Expression of CK and EpCAM can be downregulated or entirely lost during epithelial–mesenchymal transition (EMT), thus, M-CTCs can be missed by enrichment platforms that are EpCAM-based [5]. Hence, the isolation platform and the panel of markers used for CTC detection must be carefully considered to ensure accurate detection of these heterogeneous subsets of CTCs in patient’s blood.

Enrichment of CTCs has been daunting, given their extreme rarity in the circulatory system compared with WBCs. The platform used in the current study, Parsortix™, enriches CTCs based on cell size and deformability. Reports from multiple studies have demonstrated the potential of this microfluidic device to enrich CTCs with both epithelial and mesenchymal characteristics, detected either by immunocytochemistry [32] or by downstream transcripts analysis [24,33], to evaluate disease burden, progression, recurrence, and therapy responsiveness.

In a recent study employing the FQR method, the authors reported that CTCs must necessarily be positive for either of the epithelial markers, EpCAM or cytokeratin, before qualifying for analysis in the second round of re-staining [7]. We argue that important CTC subpopulations such as the M-CTCs could potentially be missed. Thus, CD45^neg^/CD16^neg^ cells should be scrutinised for mesenchymal markers in the second-round staining. We recommend the use of both CD16 and CD45 for detecting enriched WBCs, as the additive effect of these two markers has been reported to improve the staining coverage of WBCs [20], re-enforcing the accuracy and reliability of the CTC results obtained, whether E-CTCs or M-CTCs.

Moreover, additional exclusion markers should be considered when assessing CTCs. As shown in the current study, a couple of CD45^neg^/CD16^neg^ cells expressed CD31. Some of these cells would have been deemed as CTCs owing to their expression of CK/EpCAM and/or vimentin. Hence, the inclusion of vascular endothelial markers for the exclusion of circulating endothelial cells is crucial, thus reducing false positivity [13].

Our multi-marker approach allowed for efficient detection of E-CTCs, M-CTCs, and H-CTCs in the blood of HGSOC patients. E-CTCs as well as M-CTCs were found in most CTC^pos^ newly diagnosed OC patients. Notably both patients with recurrent OC also had M-CTCs. Although in a very low number of cases, this observation corresponded with a previous study result in OC, where samples from patients who underwent platinum-based chemotherapeutic regimens were found to be enriched for CTCs with mesenchymal traits [6]. The association of M-CTCs with therapy resistance has also been demonstrated previously in CTCs from breast [34] and prostate cancer patients [35].

Multiple studies have consistently demonstrated that the addition of the mesenchymal marker, N-cadherin, increases the capture rate of CTCs in OC CTC studies [13,36]. Furthermore, recent studies in OC have demonstrated the presence of mesenchymal CTCs by the expression of vimentin and their association with a poor clinical outcome [37,38].

In this study, we observed clusters of CTCs in one of the samples. Similar to an earlier report on breast cancer [34], CTC clusters were predominantly M-CTCs. Moreover, CTC clusters have been consistently demonstrated to be associated with poor patient outcome and higher metastatic potential than single CTCs in breast cancer patients [39,40]. Clusters of CTCs have also been detected in patients with OC using a conductive nano-roughened microfluidic device and were significantly associated with shorter progression-free survival (PFS) and platinum drug resistance [41].

In recent years, much attention has been given to PD-L1 because of its major role in sustaining an immunosuppressive tumour microenvironment by negatively regulating anti-tumour responses and causing anergy or exhaustion of program death receptor 1 (PD-1)-expressing T cells [42,43]. Current antibody therapies have been developed to block PD-1/PD-L1 immune interaction, to boost endogenous anti-tumour activity [44]. Anti-PD1 and anti-PDL1 targeted therapies have significantly improved the clinical outcome of patients (especially those with PD-L1-positive tumour cells, or tumour and immune cells) afflicted by various type of cancers and are currently undergoing clinical trials for patients with OC [45,46].

PD-L1 expression on CTCs from patients with multiple cancer types has been reported [47,48,49,50]. The use of different anti-PD-L1 antibody clones in these studies may explain the result discrepancies, also observed on tumour tissue staining. Thus, careful consideration was given to the selection of the 28-8 clone for our study, based on results from phase II of the Blueprint PD-L1 IHC assay (BP2) demonstrating the clinical validity and interchangeability of three different antibody clones (22C3, 28-8, and SP263) [51].

Our results revealed a significant number of PD-L1^pos^ CTCs from HGSOC patients. This study will be useful to investigate PD-L1 expression as a biomarker for inclusion in future immunotherapy clinical trials in OC as well as other cancers, to evaluate its utility to select for patients who can respond better to immunotherapeutic regimens.

Multiple studies have demonstrated the association of PD-L1 with tumours bearing a mesenchymal phenotype and its link with malignant progression [52,53,54]. Notably, in our study, PD-L1 expression was found to be associated with the H-CTCs. A similar observation was reported in a study using non-small cell lung cancer patients, where PD-L1^pos^ mesenchymal CTCs were associated with poor survival outcomes [7]. Moreover, PD-L1 expression on vascular endothelial cells has also been an area of interest in oncology [55]. For example, the expression of PD-L1 in circulating endothelial cells from patients with non-small cell lung cancer undergoing immunotherapy has been associated with poor outcomes [56], thus highlighting the idea of a potential combined anti-angiogenic agent and immunotherapy prescription for these individuals. Hence, the detection of CD31^pos^ cells expressing PD-L1 in the current study may be of relevance and warrant further scrutiny in future OC CTC studies. Type-specific antigens have been targeted for the detection of CTCs in breast cancer (HER-2) [57], melanoma (MEL and MCSP) [58], and OC (HE4) [31], and their detection rates (low to high) vary with each cancer type. The expression of OC-specific marker PAX8 was found at very low frequency on the CTCs detected in our study, although it is highly expressed in OC with serous tumours [17,18]. PAX8 is a transcription factor, and its expression levels may vary during the cell cycle. In addition, low PAX8 expression may partly be explained by the downregulation or total loss of tumour markers with time in the circulatory system, as reported earlier [11,59]. Furthermore, transcriptomic analysis has revealed that disseminated tumour cells and CTCs after reaching a new metastatic site may undergo an organ-specific mimetism and may leave that site with a new organ-specific signature [60,61]. Organ-specific mimetism has the potential to affect the expression of known markers associated with the primary tumour of origin, thus partially explaining the low PAX8 expression levels detected in the CTCs from the OC cases used in the current research.

One limitation of the study is that downstream molecular analysis was not carried out for the validation of the detected CTCs, especially those that were vimentin^pos^. Future work should involve the retrieval of these cells from the glass slides for analysis of tumour-associated genomic aberration by adapting reported methodologies [62,63]. Also, the inclusion of other markers of invasiveness, such as CD44 and E-cadherin [64], could be added to the panel for further characterisation of the CTCs in future studies.

## 5. Conclusions

Our immunostaining protocol shows the feasibility of carrying out multiple rounds of fluorescence immunostaining of CTCs on a glass slide enriched from HGSOC patients. We propose that beyond simple quantification of CTCs, future studies on OC CTCs should focus on the phenotypic heterogeneity of CTCs. Importantly, our results underscore the need to understand the biological and clinical implications of hybrid epithelial–mesenchymal CTCs in OC patients and their association with PD-L1 expression.

## Figures and Tables

**Figure 1 cancers-13-06225-f001:**
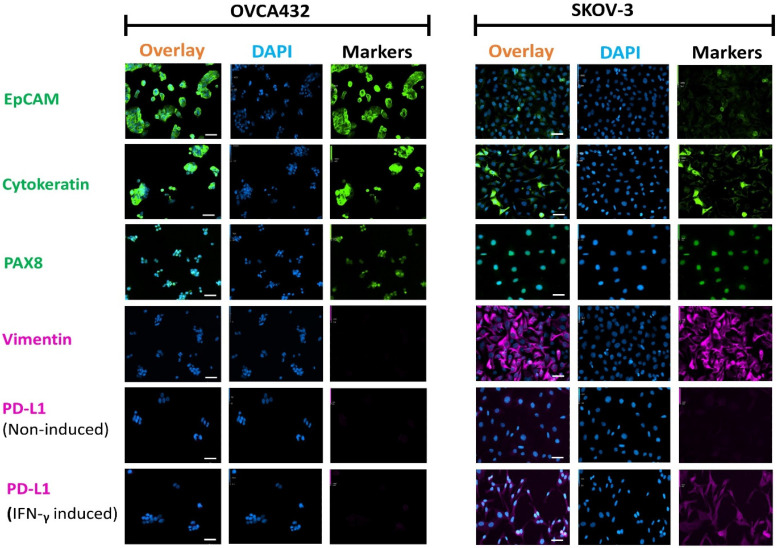
Fluorescent photomicrographs of two OC cell lines, OVCA432 and SKOV-3, stained with antibodies against EpCAM (FITC, green), Cytokeratin (FITC, green), PAX8 (AF488, green), vimentin (AF647, purple) and PD-L1 (AF647, purple). Nuclei were stained with DAPI (blue). PD-L1 expression was assessed in cells treated with IFN-γ and non-induced controls. Scale Bar = 50 µm.

**Figure 2 cancers-13-06225-f002:**
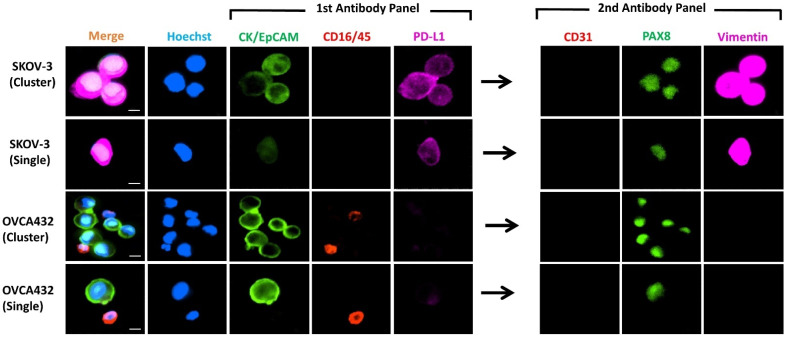
Representative images of IFN-γ-treated cell lines spiked into blood obtained from female healthy controls (*n* = 2, for each cell line). Photomicrographs show the differential expressions of the markers for CTC identification CK/EpCAM (FITC, green) and CD45/16 (PE, red), together with PD-L1 (AF647, purple) in the first antibody panel, followed by CD31 (PE, red), PAX8 (FITC, green) and vimentin (AF647, purple) in the second antibody panel post fluorescent quenching. Nuclei were stained with DAPI (blue). Merge shows overlay of channels from first and second antibody panels. Scale Bar = 10 µm.

**Figure 3 cancers-13-06225-f003:**
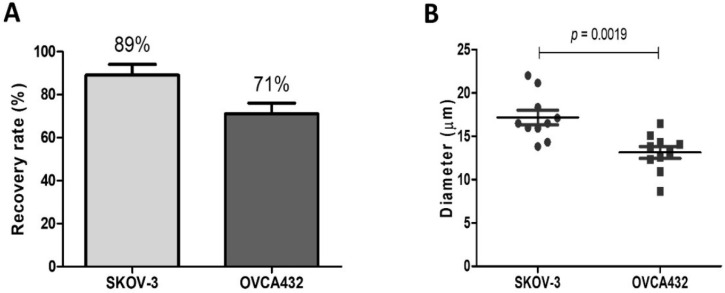
(**A**) Recovery rate of the OC cell lines after enrichment with Parsortix™ and subsequent detection by immunostaining. (**B**) Comparison of the cell diameters of the two OC cell lines (*n* = 10).

**Figure 4 cancers-13-06225-f004:**
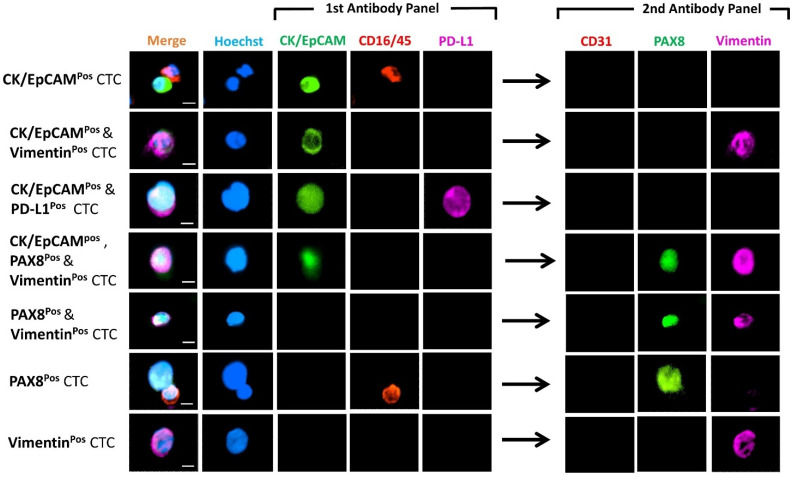
Photomicrographs of representative individual CTC phenotypes detected in the HGSOC patients. Images indicate expression of CK/EpCAM (FITC, green) and CD45/16 (PE, red), together with PD-L1 (AF647, purple) in the first antibody panel, followed by CD31 (PE, red), PAX8 (FITC, green) and vimentin (AF647, purple) in the second antibody panel post fluorescent quenching. Merge shows overlay of channels from first and second antibody panels. Nuclei were stained with DAPI (blue). Scale Bar = 10 µm.

**Figure 5 cancers-13-06225-f005:**
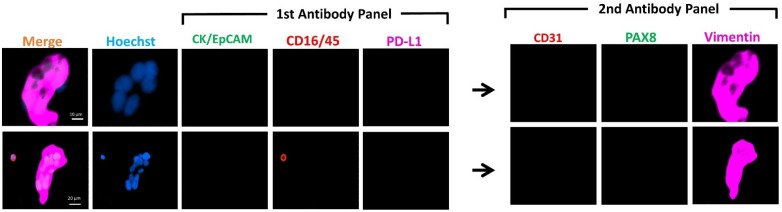
Photomicrographs showing CTC clusters expressing vimentin^pos^ (AF647, purple) in patient OC1423. Nuclei were stained with DAPI (blue). Scale Bar= 10 µm (**Upper cluster**), and 20 µm (**Lower cluster**).

**Figure 6 cancers-13-06225-f006:**
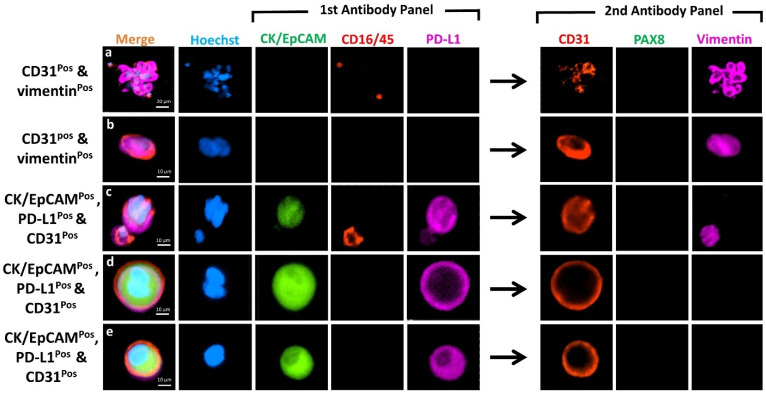
Photomicrographs showing co-expression of markers on CD31^pos^ cells in the enriched samples from the OC patients. Images indicate expression of CK/EpCAM (FITC, green) and CD45/16 (PE, red), together with PD-L1 (AF647, purple) in the first antibody panel, followed by CD31 (PE, red), PAX8 (FITC, green) and vimentin (AF647, purple) in the second antibody panel post fluorescent quenching. Merge shows overlay of channels from first and second antibody panels. Nuclei were stained with DAPI (blue). Scale = 20 µm for top row, and 10 µm for the rest.

**Figure 7 cancers-13-06225-f007:**
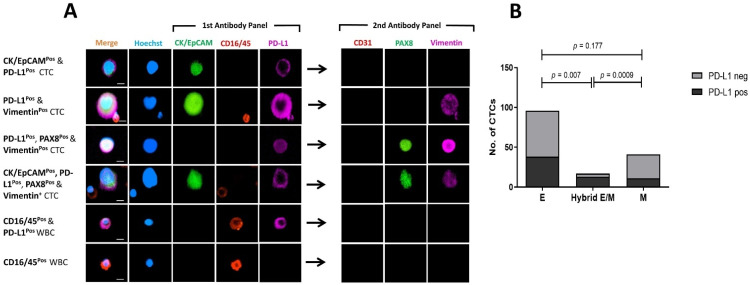
PD-L1 expression on mesenchymal and epithelial CTCs. (**A**) Photomicrograph of PD-L1^pos^ CTCs. Scale Bar = 10 µm. (**B**) Comparison of PD-L1 expression in CTCs with epithelial (E), hybrid epithelial/mesenchymal (Hybrid E/M), or mesenchymal (M) phenotype. *p*-value of Fisher’s exact test comparing E-CTCs, M-CTCs, and hybrid E/M for PD-L1 expression.

**Table 1 cancers-13-06225-t001:** Summary of clinical parameters and detected CTCs.

PID	Disease Status	Age (Years)	FIGO Stage	CA-125 (kU/L)	*BRCA* Status	Tumour PAX8	Treatment	Lymphovascular Involvement	CRS/Patient Outcome	Ascites (Volume)	CTC Count
OC 714	Recurrent	50	IIIC	114	WT	Positive	Surgery/AC	NR	NA	Absent	5
OC 1248	ND	71	IVA	354	WT	Positive	NACT	No	CRS 3	Large	0
OC 1251	ND	68	IIIC	1858	WT	Positive	NACT	Yes	CRS 2	Large	10
OC 1313	ND	55	IIIC	51	WT	Positive	NACT	Yes	CRS 1	Small	0
OC 1350	ND	72	IIIC	554	WT	Positive (Diffuse)	NACT	Yes	CRS 3	Small	0
OC 1354	ND	74	IIIC	169	WT	Positive	NACT	Yes	CRS 1	Absent	3
OC 1362	ND	68	IVA	1001	WT	Positive (Diffuse)	NACT	No	PD^1st^	Absent	8
OC 1409	ND	66	IIIC	429	Mutant	Positive	NACT	Yes	CSR 2	Small	49
OC 1418	ND	73	IVB	1200	WT	Positive	NACT	Yes	CRS 2	Medium	0
OC 1423	ND	53	IVB	313	WT	Positive (Patchy)	NACT	No	CRS 2	Small	19
OC 1382	ND	68	IC	570	WT	NR	Surgery	Yes	Good	Small	41
OC 1388	Recurrent	79	IIIA	10	WT	Positive	Surgery/AC	NR	PD^†^	NR	13
OC 1400	ND	65	IA #	27	WT	Negative	Surgery	No	Good	Absent	0
OC 1436	ND	82	IIIC	125	WT	Positive	Surgery	Yes	NA	Absent	2
OC 1458	ND	83	IVA	1056	WT	Positive	NACT	NR	PD^†a^	Small	0
OC 1364	ND	58	IIIC	576	WT	NR	NACT	NR	PD^1st^	NR	7

AC: Adjuvant chemotherapy; CRS: Chemotherapy response score; FIGO: International Federation of Gynaecology and Obstetrics; ND: Newly Diagnosed; # Mucinous borderline; NA: Not available; NR: Not reported; PD^1st^: Progressive disease with first line NACT and patient was moved out of clinical trial; PD^†^: Progressive disease after surgery and death of patient has ensued. PD^†a^: Progressive disease with patient death occurring after 3rd round of NACT, thus, CRS was not done; WT: Wild type. Criteria for volume of ascites: Large volume (ascitic fluid > 1000 mL); medium volume (between 500 and 1000 mL); small volume (<500 mL).

**Table 2 cancers-13-06225-t002:** Summary of the markers detected on the OC patients CTCs.

PID	Disease Status	CK/EpCAM	Vimentin	PAX8	Total Count (CTCs)	PD-L1
OC 714	Recurrent	2	4	3	5	0
OC 1248	ND	0	0	0	0	0
OC 1251	ND	5	7	1	10	3
OC 1313	ND	0	0	0	0	0
OC 1350	ND	0	0	0	0	0
OC 1354	ND	3	3	0	3	0
OC 1362	ND	5	3	0	8	6
OC 1409	ND	47	14	2	49	45
OC 1418	ND	0	0	0	0	0
OC 1423	ND	2	17	1	19	0
OC 1382	ND	39	3	1	41	0
OC 1388	Recurrent	3	6	1	13	7
OC 1400	ND	0	0	0	0	0
OC 1436	ND	2	0	0	2	0
OC 1458	ND	0	0	0	0	0
OC 1364	ND	5	1	2	7	1
**TOTAL**		113	58	11	157	62

ND: Newly diagnosed.

## Data Availability

All relevant data has been presented in the article. Raw images of reported results have been stored at the School of Medical and Health Sciences and at the Centre for Precision Health, Edith Cowan University.

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
