# Peer review of "Multi-Marker Immunofluorescent Staining and PD-L1 Detection on Circulating Tumour Cells from Ovarian Cancer Patients"

_cancers, 2021, doi:10.3390/cancers13246225_

Round 1
Reviewer 1 Report
There are several comments on the material submitted for publication:
1. СТС is heterogeneous cells, but some important markers such as CD44, E-cadherin and the like are not included in the publication.
2. There are grounds for selecting the presented ovarian cancer cell lines for experimentation?
Author Response
Thank you for the time taken to review our manuscript and your feedback.
Please see below the response to your comments and edits inserted in the revised manuscript.
There are several comments on the material submitted for publication:
1. СТС is heterogeneous cells, but some important markers such as CD44, E-cadherin and the like are not included in the publication.
Response: The reviewer is correct that we did not include these markers in this study. However, our aim was to increase markers that would enhance the detection of CTCs, rather than an extended characterisation. Our immune staining protocol made room for only five (minus DAPI) markers for detection of CTC and PD-L1. The other markers you mentioned are beyond the scope of this manuscript but would be good to pursue in future studies.
We added a referenced sentence to the discussion section in line (480), highlighting your suggestion, and it states: “Also, the inclusion of other markers of invasiveness, such as CD44 and E-cadherin [64], could be added to the panel for further characterisation of the CTCs in future studies.”
There are grounds for selecting the presented ovarian cancer cell lines for experimentation?
Response: We selected these two ovarian cancer cell lines based on their phenotypic characteristics. The SKOV-3, a more mesenchymal cell line for the demonstration of the mesenchymal marker, vimentin, and a OVCA432 which is more epithelial cell line, with very high expression of the epithelial markers, and no expression of the mesenchymal marker (vimentin).
This was explained in our manuscript (line 214-218) and has also been referenced. It states: “The more cluster forming cell line, OVCA432, showed high expression of the epithelial markers EpCAM and CK, while the less cluster forming and larger cell line, SKOV-3, generally demonstrated very weak expression for EpCAM and strong expression of the mesenchymal marker vimentin [22].”
Additional changes:
- Please note that there was an error in the numbers inputted in the analysis of E-, M-CTCs and H-CTCs (Figure 7B), where we fail to discount the hybrid CTCs from the E- and M-CTCs counts. We have revised the Figure with the corrected statistics and the associated numbers in the text in line 351-354): “PD-L1pos/CTCs accounted for 11/41 (27%) M-CTCs and 38/96 (40%) E-CTCs (Fisher’s exact test, p=0.177). PD-L1pos CTCs were more common among H-CTCs, representing 13/17 (76%) of these cells. Positivity for PD-L1 was significantly associated with H-CTCs when compared with epithelial (p=0.007) and mesenchymal (p=0.0009) CTCs (Fisher’s exact test).”
- The Centre for Precision Health was added as an additional affiliation for a few of the authors.
Reviewer 2 Report
The aim of this study is to analyze the microfluidic enrichment of ovarian circulating tumour cells (CTCs), providing a multi-marker staining methodology for the identification of heterogeneous CTCs in ovarian cancer (OC) patients, with an organic evaluation of PD-L1 expression.
Even if the manuscript provides an organic overview, with a densely organized structure and based on well-synthetized data, there are aspects to be mentioned, to make the article fully readable. For these reasons, the manuscript requires minor changes.
Please find below an enumerated list of comments on my review of the manuscript:
LINE 184: In this section, the manuscript may benefit from providing a brief and complete histological classification of these subtypes of ovarian cancer, specifically in terms of their morpho-functional features (see, for reference: Giusti, I.; Bianchi, S.; Nottola, S.A.; Macchiarelli, G.; Dolo, V. Clinical electron microscopy in the study of human ovarian tissues. EuroMediterr. Biomed. J. 2019, 14, 145–151), in order to complete this information.
MATERIAL AND METHODS:
As regards this section, the methodology design was rigorous and appropriately implemented within the study.
RESULTS:
Also this section is well organized and densely presented, based on well-synthetized data.
DISCUSSION:
LINE 439: This lineage – specific transcription factor is over – expressed in several reproductive malignancies, exerting a pivotal role as a supporter of the survival and proliferation of tumor cells (see, for reference: Chaves-Moreira, D.; Morin, P. J.; Drapkin, R. Unraveling the mysteries of PAX8 in reproductive tract cancers. Cancer Research. 2021, 81(4), 806-810.
In conclusion, this manuscript is densely presented and well organized, based on well-synthetized data. The authors were lucid in their style of writing, making it easy to read and understand the message, portrayed in the manuscript. However, many of the topics are very concisely covered. Moreover, this research have futuristic importance and could be potential for future research. However, I have minor comments only for the introductive and discussion section, for improvement before acceptance for publication. I would accept the manuscript, if the comments are addressed properly.
Author Response
Thank you for the time taken to review our manuscript and your feedback.
Please see below the response to your comments and edits inserted in the revised manuscript.
- LINE 184: In this section, the manuscript may benefit from providing a brief and complete histological classification of these subtypes of ovarian cancer, specifically in terms of their morpho-functional features (see, for reference: Giusti, I.; Bianchi, S.; Nottola, S.A.; Macchiarelli, G.; Dolo, V. Clinical electron microscopy in the study of human ovarian tissues. EuroMediterr. Biomed. J.2019, 14, 145–151), in order to complete this information.
Response: We have inserted the suggested comments in the introduction section (line 82-88), of the manuscript, and it reads, “Ovarian cancer (OC) are histologically heterogeneous, with epithelial ovarian carcinoma making up about 90% of all cases of the disease. Epithelial ovarian carcinoma consist of four distinct histological subtypes, namely, clear cell, serous, mucinous and endometrioid. The high grade serous ovarian carcinoma (HGSOC) subtype, is the most clinically aggressive and common form of the disease [15]. Thus, techniques that enhance efficient detection of CTCs with markers of biological importance in this lethal OC subtype is warranted.”
- MATERIAL AND METHODS:
As regards this section, the methodology design was rigorous and appropriately implemented within the study.
Response: We thank the reviewer for this comment
- RESULTS:
Also this section is well organized and densely presented, based on well-synthetized data.
Response: We thank the reviewer for this comment
- DISCUSSION:
LINE 439: This lineage – specific transcription factor is over – expressed in several reproductive malignancies, exerting a pivotal role as a supporter of the survival and proliferation of tumor cells (see, for reference: Chaves-Moreira, D.; Morin, P. J.; Drapkin, R. Unraveling the mysteries of PAX8 in reproductive tract cancers. Cancer Research. 2021, 81(4), 806-810.
Response: The suggested article above has been added to a sentence (lines 468-470) in the manuscript at the discussion section, as suggested by the reviewer, and it states: “PAX8 is a transcription factor, and its expression levels may vary during the cell cycle. In addition, PAX8 low expression may partly be explained by the downregulation or total loss of tumour markers with time in the circulatory system, as reported earlier [11, 58].”
Additional changes:
- Please note that there was an error in the numbers inputted in the analysis of E-, M-CTCs and H-CTCs (Figure 7B), where we fail to discount the hybrid CTCs from the E- and M-CTCs counts. We have revised the Figure with the corrected statistics and the associated numbers in the text in line 351-354): “PD-L1pos/CTCs accounted for 11/41 (27%) M-CTCs and 38/96 (40%) E-CTCs (Fisher’s exact test, p=0.177). PD-L1pos CTCs were more common among H-CTCs, representing 13/17 (76%) of these cells. Positivity for PD-L1 was significantly associated with H-CTCs when compared with epithelial (p=0.007) and mesenchymal (p=0.0009) CTCs (Fisher’s exact test).”
- The Centre for Precision Health was added as an additional affiliation for a few of the authors.